# Activated Carbon and Clay Pellets Coated with Hydroxyapatite for Heavy Metal Removal: Characterization, Adsorption, and Regeneration

**DOI:** 10.3390/ma16093605

**Published:** 2023-05-08

**Authors:** Inga Jurgelane, Janis Locs

**Affiliations:** 1Rudolfs Cimdins Riga Biomaterials Innovations and Development Centre of RTU, Institute of General Chemical Engineering, Faculty of Materials Science and Applied Chemistry, Riga Technical University, Pulka 3, LV-1007 Riga, Latvia; janis.locs@rtu.lv; 2Baltic Biomaterials Centre of Excellence, Headquarters at Riga Technical University, LV-1048 Riga, Latvia

**Keywords:** activated carbon, hydroxyapatite, pellets, heavy metals, adsorption, regeneration

## Abstract

In the present work, activated-carbon-containing pellets were preparedby direct chemical activation of sawdust, using clays as a binder. The obtained pellets (ACC) were coated with hydroxyapatite (HAp) nanoparticles (ACC-HAp) to improve adsorption towards Pb(II), Cu(II), Zn(II), and Ni(II). The pellets were characterized by scanning electron microscopy (SEM), by Fourier transform infrared spectroscopy (FTIR), and with a gas sorptometer. The effect of pH, contact time, and initial concentration on adsorption performance was investigated. Additionally, desorption studies were performed, and the regeneration influence on compressive strength and repeated Pb(II) adsorption was investigated. The results showed that, after coating ACC pellets with HAp nanoparticles, the adsorption capacity increased for all applied heavy metal ions. Pb(II) was adsorbed the most, and the best results were achieved at pH 6. The adsorption process followed the pseudo-second-order kinetic model. The adsorption isotherm of Pb(II) is better fitted to the Langmuir model, showing the maximum adsorption capacity of 56 and 47 mg/g by ACC-HAp and ACC pellets, respectively. The desorption efficiency of Pb(II)-loaded ACC-HAp pellets increased by lowering the pH of the acid, resulting in the dissolution of the HAp coating. The best desorption results were achieved with HCl at pH 1 and 1.5. Therefore, the regeneration procedure consisted of desorption, rinsing with distilled water, and re-coating with HAp nanoparticles. After the regeneration process, the Pb(II) adsorption was not affected. However, the desorption stage within the regeneration process decreased the compressive strength of the pellets.

## 1. Introduction

Water and wastewater pollution with heavy metals is a global problem. Heavy metal pollution can originate from both natural and anthropogenic sources. The main natural sources are weathering of rocks and volcanic activities. Some of the primary anthropogenic sources are mining, mineral processing, agriculture, and manufacturing. Most heavy metals are toxic and carcinogenic and, therefore, can cause serious harm to the environment and human health. They can damage the liver, kidney, brain, nervous system, and blood; cause Alzheimer’s disease; and be lethal even at low concentrations. For example, excess copper retention causes Wilson’s disease, and copper and zinc impede neurodevelopment when excessive amounts enter the brain. Nickel works as a carcinogen by controlling carcinogenic mechanisms, but lead has a deleterious effect on all organs, especially kidneys [1,2,3].

There are many methods for removing heavy metals from wastewater—chemical precipitation, ion exchange, adsorption, reverse osmosis, coagulation and flocculation, membrane separation, etc. Among these methods, adsorption is considered the most effective and economically viable. The adsorption process also has a regeneration possibility for the adsorbent, thereby recovering the heavy metals as they cannot be destroyed or degraded [2,4,5].

The properties of an ideal heavy metal adsorbent are high adsorption capacity, ease of handling in dynamic conditions, mechanical durability (for granules and pellets), reusability after regeneration, and high selectivity for better heavy metal recovery. The adsorbent also should have a long application life and, afterwards, have the possibility for easy recycling or discarding without a negative impact on the environment. Low cost and uncomplicated manufacturing are also important factors. In the last few decades, various adsorption materials have been investigated—natural and synthetic, inorganic and organic, pure materials, and in combination with other materials (composites) [4,5]. Activated carbon (AC) is a widely studied material for removing heavy metals due to its high specific surface area and effective adsorption properties. AC has relatively high affinity towards Pb(II), Cd(II), Cu(II), Ni(II), and Zn(II) [6,7,8,9]. Application of AC in powder form in the adsorption process has some disadvantages, such as agglomeration and filtration difficulties due to very fine particles that form a dense filter layer in slurry phase operations [10]. Additionally, a huge volume of adsorbent is required in the adsorption process. In closed vessels, high hydraulic pressure and flow resistance can occur, and, afterward, the used fine particle adsorbent is difficult to operate and regenerate [11]. Therefore, applying AC in the form of pellets or granules would be essential to avoid these disadvantages.

AC is prepared from various raw materials, but the most common are wood, charcoal, coconut, peat shells, lignite, and biomass [12]. The application of certain materials for AC preparation and investigation-related studies is probably based on these materials’ availability in the surrounding area or country. Timber processing is Latvia’s most significant industrial sector because forests cover more than 50% of the territory [13]. The sawdust generated from wood processing is used to manufacture products like pellets, briquettes, and particle boards and in various scientific research projects to develop new products. Nevertheless, a lot of sawdust waste is just discarded, so developing a new product would increase the practical application of this material.

Activated carbon can be prepared through chemical and physical activation. Although physical activation is a chemical-free process, it has two main disadvantages—long activation time and high energy consumption [11]. In the chemical activation of sawdust, the AC can be prepared through a direct activation [11,14,15] or a two-stage process including initial carbonization or pyrolysis and then activation [12,16,17]. The direct activation process is less time-consuming and low-cost and effective adsorption properties can be achieved. In the literature, most research on adsorption is conducted AC obtained from sawdust as a powder or small particles (wood chips). There is a lack of information on AC pellets or granules prepared from sawdust for heavy metal adsorption. Tang et al. [11] investigated malachite green dye removal with AC pellets prepared by mixing AC and acid-impregnated sawdust, hand-pressing in a cylindrical mold, and pyrolyzing. Li et al. [14] investigated the physical characteristics of AC pellets prepared by direct activation of acid-impregnated sawdust pressed as pellets. Therefore, in this study, AC pellets were prepared via direct activation of an extruded mixture of sawdust, acid, and clays. Clays were added as a binder due to their inertness and thermal resistance.

There are numerous studies about the modification of AC to improve the adsorption capacity and affinity to specific contaminants [18,19]. One of the materials used in AC modification is hydroxyapatite (HAp, chemical formula Ca_10_(PO_4_)_6_(OH)_2_). HAp is the most stable calcium phosphate form, naturally occurring in bones and teeth. Synthesized HAp is mainly used in bone defect regeneration [20]. However, there are studies about applying HAp as a macronutrient nano-fertilizer in agriculture [21] and in the defluoridation method (fluoride removal) for drinking water [22]. HAp can also be used for the removal of heavy metal ions like Pb(II), Zn(II), Cu(II), and Ni(II) from aqueous solutions [23,24,25,26,27]. Jayaweera et al. [28], Fernando et al. [29], and Long et al. [30] showed that HAp and granular AC (GAC) composites have higher Pb(II) adsorption than pure GAC. Studies by Wang et al. [7] and Chen et al. [31] revealed as well that HAp and biochar (BC) composites have increased adsorption towards Pb(II), Cu(II), Zn(II), and Cd(II) compared to pure BC. In these previous studies, HAp composites were prepared using an in situ method—during HAp synthesis. In this study, HAp nanoparticles are coated using the dip coating method—by immersing AC pellets into diluted HAp particle solution.

The aim of this work is to prepare AC pellets by direct chemical activation of sawdust, using clay as a binder, and to modify them with HAp nanoparticles to improve adsorption properties towards the following heavy metals: Pb(II), Cu(II), Zn(II), and Ni(II). The obtained pellets were characterized with various methods; Pb(II) adsorption was conducted initially and the possible regeneration process was investigated.

## 2. Materials and Methods

### 2.1. Materials

Pine sawdust with a particle size of <0.125 mm was obtained from the Latvian State Institute of Wood Chemistry. Illitic clays were obtained in the southwest of Latvia at 2–2.5 m depth. A clay fraction with particle size < 63 µm (obtained by wet sieving) containing 52% clay minerals was used. For the chemical activation of sawdust, concentrated H_2_SO_4_ (Trace Metal Grade, Sigma Aldrich, St. Louis, MO, USA) was used. For the synthesis of HAp, ≥97.0% (from marble) CaO (Fluka, Charlotte, NC, USA) and 85% H_3_PO_4_ (Sigma Aldrich, St. Louis, MO, USA) were used. For adsorption and desorption experiments, analytical-grade NaCl, NaOH, HCl, Pb(NO_3_)_2_ and Ca(II), Mg(II), Pb(II), Cu(II), Ni(II), and Zn(II) standard solutions with concentrations of 1000 mg L^−1^ (Sigma Aldrich, St. Louis, MO, USA) were used.

### 2.2. Preparation of ACC Pellets

Sawdust was mixed with clays and 50% sulfuric acid. The volume ratio of sawdust, clays, and sulfuric acid was 1:1:1, accordingly. The mass was mechanically mixed and extruded through a syringe with a diameter of 4 mm. The obtained rods were dried on a stove at 100 °C for 30–40 min, broken into 6–8 mm-long pieces, and pyrolyzed at 500 °C for 2 h with a heating rate of 5 °C min^−1^ in nitrogen flow. After cooling, the pellets were rinsed to remove the remaining sulfate ions.

### 2.3. Synthesis of HAp

HAp was synthesized by a wet precipitation reaction between Ca(OH)_2_ and H_3_PO_4_, as described before [32], with few modifications. The suspension of 0.15 M Ca(OH)_2_ was obtained by the “lime slaking “process, where CaO is vigorously mixed with distilled water for 1 h. Then, 2 M H_3_PO_4_ was slowly (~0.75 mL/min) added to the Ca(OH)_2_ suspension under vigorous stirring. The temperature of the synthesis was maintained constant at 45 °C. When the pH reached 7, the synthesis solution was stabilized (stirred) for 2 h. The obtained precipitates were matured at ambient temperature for 14–15 h and then filtered.

### 2.4. HAp Coating

Next, 2–3 wt% HAp nanoparticle suspension was prepared from filtered HAp precipitates. The pellets were immersed in HAp suspension for 20–30 s, taken out, and dried at 60 °C until reaching constant mass. This procedure was repeated two times.

### 2.5. Characterization of Pellets

Specific surface area (SSA), pore size distribution, and total pore volume were determined with the nitrogen gas adsorption method, performed with QuadraSorb SI (Quantachrome Instruments, Boynton Beach, FL, USA). SSA was calculated according to the BET method. The pore size distribution and total pore volume were derived from desorption branches of isotherms using the BJH model.

The point of zero charge (pH_pzc_) was determined using the pH drift method. One pellet (~0.05–0.06 g) was immersed in 20 mL of 0.01 M NaCl solution with pH values between 3 and 10. After 24 h under agitation, the final pH was measured. The pH_pzc_ is the point where pH_initial_ − pH_final_ = 0.

Surface morphology was investigated with a scanning electron microscope (SEM) Phenom ProX (PhenomWorld, Eindhoven, The Netherlands). The acceleration voltage was 5 kV.

For the analysis of functional groups, Fourier transform infrared (FTIR) spectra were taken using a Thermo Scientific Nicolet™ iSTM50 (Thermo Fisher Scientific, Waltham, MA, USA) spectrometer in the Attenuated Total Reflectance (ATR) mode. Spectra were obtained over a range of wavenumbers from 400 cm^−1^ to 4000 cm^−1^, using 64 scans with 4 cm^−1^ resolutions. For the measurements, the pellets were crushed in a pestle. Before every measurement, a background spectrum was taken.

Compressive strength tests were conducted with a testing machine Tinius Olsen 25ST (Tinius Olsen, Redhill, UK), using a load cell with a maximum load capacity of 250 N and a compression speed of 1 mm/min. The pellets were compressed vertically.

### 2.6. Adsorption Studies

For all experiments, one pellet (~ 0.05–0.06 g) was mixed with 5 mL of heavy metal solution and shaken at 150 rpm at room temperature. Multi-metal solutions containing Cu(II), Pb(II), Ni(II), and Zn(II) (50 mg L^−1^ each) were used for sorption kinetics and pH influence on adsorption. These solutions, with pH 4–6 and an adsorption time of 24 h, were used to evaluate the pH influence on removal percentage. Multi-metal solution with pH 6 at various adsorption time intervals (8, 16, 20, 24, 30, and 40 h) was examined for sorption kinetics experiments. The pH of the solutions was adjusted with HCl and NaOH. Pb(II) solutions with pH 6 and various initial concentrations (50–1500 mg/L) were used to obtain sorption isotherms. All used solutions for adsorption experiments also contained background cations—80 mg L^−1^ of Ca(II) and 30 mg L^−1^ of Mg(II). These ions were added to bring the experiment closer to actual adsorption conditions because all types of water (except for distilled) contain calcium and magnesium ions, and they can decrease the adsorption efficiency. Heavy metal concentrations were determined using an inductively coupled plasma mass spectrometer ICP-MS, 8900 Triple Quad (Agilent Technologies St. Clair, CA, USA). The removal percentage (R %) was calculated as follows:(1)$$R\;\%=\left(C_{0}-C_{e}\right)/C_{0}*100\%$$
where $C_{0}$ and $C_{e}$ (mg/L) are the initial and equilibrium heavy metal concentrations, respectively.

### 2.7. Adsorption Kinetic and Equilibrium Models

To analyze and describe the kinetics of Pb(II), Cu(II), Zn(II), and Ni(II) adsorption, pseudo-first-order (PFO) and pseudo-second-order (PSO) models were used [33]. The linear equations for PFO and PSO models are expressed by the following Equations (2) and (3), accordingly:(2)$$ln\left(Q_{e}-Q_{t}\right)=lnQ_{e}-k_{1}t$$
(3)$$t/Q_{t}=1/k_{2}Q_{e}{}^{2}+1/Q_{e}*t$$
where $Q_{t}$ (mg/g) is the adsorbed amount at a certain time (*t*); $Q_{e}$ is the adsorbed amount (mg/g) at equilibrium; and *k*_1_ (h^−1^) and *k*_2_ (g/mg·h) are the adsorption rate constants of PFO and PSO, respectively.

Langmuir and Freundlich isotherm models [34,35] were applied for Pb(II) adsorption to interpret the adsorption isotherm data. The linear form of the Langmuir isotherm is as follows:(4)$$C_{e}/Q_{e}=1/q_{max}*K_{L}+C_{e}/q_{max}$$
where $C_{e}$ is the metal ion concentration (mg/L) at equilibrium, $K_{L}$ is the Langmuir constant (L/mg), and $q_{max}$ is the maximum adsorption capacity (mg/g). The linear form of the Freundlich isotherm is as follows:(5)$$logQ_{e}=logK_{F}+\frac{1}{n}*logC_{e}$$
where $K_{F}$ is the adsorption capacity (mg/g) and *n* is the adsorption intensity.

### 2.8. Desorption Studies

One ACC-HAp pellet was mixed with 5 mL of Pb(II) solution with a concentration of 50 mg/L and pH 6 and shaken at 150 rpm at room temperature. After 20 h, the metal-loaded pellet was removed, slightly rinsed with distilled water, and inserted into a flask with 5 mL of HCl or HNO_3_ solutions at pH values of 1–3 for 30 min. Afterward, the pellet was rinsed with distilled water until reaching neutral pH. Desorption efficiency (*D*%) was calculated as follows:(6)$$D\;\%=m_{des}/m_{ads}*100\%$$
where *m_ads_* and *m_des_* (mg) are the masses of Pb(II) adsorbed on the adsorbent and desorbed from the adsorbent, respectively.

### 2.9. Regeneration and Repeated Adsorption

One ACC-HAp pellet was mixed with 5 mL of 200 mg/L Pb(II) solution at pH 6 and shaken at 150 rpm at room temperature. After 20 h, the pellet was removed and the residue Pb(II) concentration was measured in the solution. The pellet was slightly rinsed with distilled water and inserted into a flask with 5 mL of HCl (1) at pH 1–2 for 30 min for compressive strength experiments and (2) at pH 1.5 for 1 h for repeated adsorption. Afterward, the pellet was rinsed with distilled water until reaching neutral pH, dried, and re-coated with HAp particles as described above. This adsorption–regeneration cycle mentioned above was repeated three more times.

## 3. Results and Discussion

### 3.1. Physical Characterization

The physical appearance of both pellets can be seen in Figure 1. ACC pellets are black but become greyish/blue when coated with HAp nanoparticles. The morphology of the ACC-HAp pellet is shown in Figure 2. Clay mineral particles (sheets) can be seen inside the pellet (Figure 2a). Figure 2b shows the surface of the ACC-HAp pellet.

The pore diameter for ACC pellets is mainly centered at about 8 nm, as obtained from the pore size distribution (Figure 3), whereas ACC-HAp pellets contain a slightly larger amount of smaller pores than ACC with a peak at about 4 and 5.5 nm. As shown in Table 1, ACC pellets have slightly larger total pore volume and SSA than ACC-HAp—0.41 and 0.37 cm^3^/g and 188 and 195 m^2^/g, respectively. However, the statistical evaluation, using an unpaired Student’s *t*-test (significance level set at *p* < 0.05), shows no statistically significant difference in these characteristics between ACC and ACC-HAp pellets.

The pH_pzc_ for ACC is 6.3, but for ACC-HAp, 6.5. Since the adsorption processes are conducted at lower pH than pH_pzc_, both pellets have a positive net surface charge during the adsorption, which is unfavorable for the adsorption of positively charged metal ions.

The FTIR spectra of ACC pellets and ACC-HAp pellets before and after adsorption of Pb(II), Cu(II), Ni(II), and Zn(II) are shown in Figure 4. The broad band at around 3400 cm^−1^ can be assigned to O-H stretching, but the one at around 580 cm^−1^ to O-H bending vibrations [36]. The band at 1000–1100 cm^−1^ corresponds to the vibrations of C-O [37]. The band around 1600 cm^−1^ is assigned to O-H bending from the absorbed water. The bands at 798 cm^−1^, 778 cm^−1^, and 692 cm^−1^ are attributed to Si-O stretching from quartz particles in clay. The band at around 400–460 cm^−1^ could be attributed to Si-O bending from clays [38]. In the ACC-HAp spectra, an additional band at 564 cm^−1^ corresponds to the vibrations of the phosphate groups from the coated HAp particles. The presence of the other HAp phosphate group around 1037 cm^−1^ can be seen as a slight shift to the right of the band at 1000–1100 cm^−1^ [30]. After the adsorption, the intensity of the band at 1000–1100 cm^−1^ decreases, suggesting that -PO_4_ and C-O functional groups were involved in the adsorption process.

### 3.2. Adsorption Studies

The effect of initial pH on the competitive adsorption of Pb(II), Cu(II), Ni(II), and Zn(II) by both ACC and ACC-HAp pellets are shown in Figure 5. The results show that the uptake percentage increases by increasing the pH, and the maximum adsorption of all ions is reached at pH 6 for both pellets. Therefore, in this work, pH 6 was used in other adsorption studies as the optimum pH value, similar to other studies in the literature [7,28,29].

The influence of contact time on the competitive adsorption of heavy metals by ACC and ACC-HAp is shown in Figure 6. According to the results, Pb(II) is being adsorbed at a greater extent and faster than other ions. In the first 8 h, ACC-HAp removes 76% from the added Pb(II) concentration, which is 15% more than ACC. Overall, most of the heavy metals are adsorbed in the first 8 h due to the large availability of functional groups on the surface. The filling of these groups results in a slower further adsorption process. When the adsorption equilibrium for Pb(II) and Cu(II) is reached (at 20 and 24 h, respectively), the difference between the removal percentage of these two ions has decreased. For Ni(II) and Zn(II), the adsorption equilibrium is achieved between 24 and 30 h.

According to Figure 5 and Figure 6, ACC-HAp has slightly higher adsorption than ACC towards all ions. Similar results were obtained in other studies, where Pb(II), Cu(II), and Zn(II) adsorption was improved after modifying GAC [28,29] and BC [7,31] with HAp particles.

All pellets have much higher adsorption towards Pb(II) and Cu(II) than towards Zn(II) and Ni(II). Overall, the highest removal percentage is for Pb(II) for both pellet types. Other studies about competitive adsorption that included Pb(II), Cu(II), and other ions with both AC [1,39,40] and HAp [41,42] showed that Pb(II) is adsorbed better than other ions like Cu(II), Cd(II) and Ni(II). It is suggested that this could be influenced mainly by a combination of three factors—electronegativity, radius, and free energy of hydrated ions [42,43,44]. A higher electronegativity has a positive effect on adsorption efficiency, and for Pb(II), it is 2.33 (Paulig), but for Cu(II), Ni(II), and Zn(II), it is 1.9, 1.91, and 1.65, respectively [42,43,45]. For ions with a smaller hydrated radius, the center of the charge is closer to the surface of the adsorbent, and the electrostatic interaction is stronger. Ions with lower hydration energy are easier to dehydrate and thus adsorb. The radius and energy of hydrated Pb ions are smaller than for other ions—0.401 nm and −1481 kJ/mol, respectively. For Cu(II), Ni(II), and Zn(II), the hydrated radius is 0.419, 0.404, and 0.430 nm, respectively, but the hydration energy is −2100, −2106, and −2046 kJ/mol, respectively [43,44]. For HAp, the favorable adsorption towards Pb(II) is also explained by the ionic radius of 1.19 Å, which is close to the Ca(II) radius of 0.99 Å [42,44]. For Cu(II), Ni(II), and Zn(II), the ionic radius is 0.73, 0.69, and 0.74 Å, respectively [42].

To determine the rate-controlling step of the adsorption process, linear regression of the experimental data with two kinetic models was performed. The graphical plots of the PSO and PFO models for the competitive adsorption of Pb(II), Cu(II), Zn(II), and Ni(II) are shown in Figure 7. The obtained kinetic parameters and correlation coefficients (R^2^) were calculated and listed in Table 2. The visual analysis of the graphical plot suggests that the PSO model better fits the experimental data. It is approved by the regression results, where R^2^ values for the PSO model are higher than those of the PFO kinetic model. Moreover, the calculated (*Q_e, cal_*) values of the PSO model are much closer to the experimental (*Q_e, exp_*) values than for the PFO model. For Pb(II) and Cu(II), the *Q_e, cal_* values are more similar to those of *Q_e, exp_* than for Ni(II) and Zn(II). The obtained results indicate that the adsorption process is controlled by chemical reactions, such as ion exchange, surface complexation, and/or precipitation [26,33]. These reactions can occur on AC and HAp surfaces [26,27,46]. This rate-controlling step agrees with the fact that both pellets have a net positive charge during adsorption due to the pH_pzc_ values (Table 1). Therefore, the adsorption is not controlled by electrostatic attraction. Other studies on heavy metal adsorption by activated carbon and HAp composite also showed that the PSO model best fits the experimental data [7,30,31].

The effect of the initial Pb(II) concentration on ACC and ACC-HAp adsorption capacity was investigated. The obtained data are interpreted and plotted using the Langmuir and Freundlich models (Figure 8). The constants of both models in Table 3 are calculated using the slope and intercept of the linear trend lines of graphs $C_{e}$/$Q_{e}\;$vs. $C_{e}$ and log $Q_{e}$ vs. log$\;C_{e}$, respectively.

According to the R^2^ values, ACC and ACC-HAp isotherm experimental data are better fitted by the Langmuir model (0.9866 and 0.9969) than the Freundlich model (0.9274 and 0.9714), suggesting that Pb(II) forms a monolayer coverage [47] on these pellets. Previous research showed that AC and AC and HAp composites can form a monolayer and multilayer coverage [28,29].

ACC-HAp pellets have a higher K_L_ value than ACC, indicating higher affinity towards Pb(II), confirmed by ACC-HAp’s higher maximum adsorption capacity (*q_max_*) than ACC pellets—56 and 47 mg/g, accordingly. Such results could be explained as due to slightly higher SSA for ACC-HAp than for AC, and many studies showed that HAp particles have a higher affinity towards Pb(II) ions than AC [6,48,49].

The adsorption performance was compared with other modified and non-modified AC adsorbents in granular and pellet forms in the literature and summarized in Table 4. Both ACC and ACC-HAp show good and competitive Pb(II) adsorption performance, considering the presence of calcium and magnesium ions mentioned in Section 2.6. For example, GAC from coconut shells studied by Caccin et al. [50] showed 1.7 times higher Pb(II) adsorption, but took two times longer to achieve equilibrium time than ACC-HAp. Jayaweera et al.’s [28] study revealed that GAC modification with HAp particles increased the adsorption capacity by 18.7%. However, in this study, the adsorption capacity of pellets coated with HAp particles was increased to only slightly higher—by 19.1%. Nevertheless, the added value of ACC-HAp pellets is the application of the dip coating instead of the in situ method for HAp coating, which is more time-consuming and complicated.

### 3.3. Desorption Studies

The desorption process is essential for reusing the adsorbent. In the desorption process of AC, usually, inorganic acids like HCl, HNO_3_, and H_2_SO_4_ are used [9,43,54,55], but there are scarce studies about HAp regeneration possibilities. Wang et al. [7] used 0.2 M HCl (pH = 0.7) for Pb(II) desorption from BC and HAp composite.

In this study, HCl and HNO_3_ were used at a pH range of 1–3 for ACC-HAp regeneration studies, as shown in Figure 9. The results show that the desorption efficiency is decreased by increasing the pH value. For both acids, there is a rapid decrease in desorption efficiency at pH > 2. HCl shows the highest desorption efficiency—98% and 97% at pH 1 and 1.5, respectively. It was visually observed that, at a pH range of 1–2, HAp particles were fully dissolved from the ACC-HAp pellet surface, but HAp was dissolved partially at pH > 2. These HAp dissolution observations positively correlate with desorption efficiency, suggesting that Pb(II) primarily adsorbs onto the HAp surface. On the other hand, in AC desorption, the inorganic acids are used at pH < 1.1 [43,54,55], suggesting that desorption efficiency decreases at higher pH.

### 3.4. Compressive Strength

In addition to sufficient adsorption properties, the adsorbents must also be mechanically durable for better handling of the material. The compressive strength of untreated individual ACC-HAp pellets and those pellets after four adsorption–regeneration cycles with HCl solution at pH 1–2 can be seen in Figure 10. The results show that the compressive strength of untreated ACC-HAp pellets varies from 0.93–1.3 MPa, with the average value of 1.1 ± 0.1 MPa. After four adsorption–regeneration cycles, the compressive strength decreased. The most rapid decrease is after desorption at pH 1, where the average compressive strength is 0.66 ± 0.04 MPa. At pH 1.5 and 2, the average compressive strength is similar—0.88 ± 0.05 MPa and 0.92 ± 0.09 MPa, respectively. Due to these and the Section 3.3 results, desorption with HCl at pH 1.5 was chosen for regeneration and repeated adsorption experiments.

### 3.5. Regeneration Influence on Pb(II) Adsorption

To evaluate the reusability of ACC-HAp pellets, regeneration and repeated adsorption were performed. The adsorption capacity in the first adsorption–regeneration cycle (Figure 11) is before the regeneration process. The results show that the adsorption capacity in every following cycle is different—both higher and lower than in the first cycle. This is probably because each regeneration process involves coating new HAp particles that restore the adsorption properties. The adsorption capacity slightly varies between the pellets because of uneven HAp coating that can be observed visually. However, based on the statistical evaluation, there is no statistically significant difference in the adsorption capacity between these cycles.

## 4. Conclusions

In this study, activated carbon and clay pellets (ACC) were prepared by direct chemical activation of sawdust and clay pellets. These pellets were successfully coated with hydroxyapatite (HAp) nanoparticles by dipping them into dilute HAp suspension. The obtained ACC-HAp pellets exhibited higher adsorption towards Pb(II), Cu(II), Zn(II), and Ni(II) than ACC. The adsorption of all ions by both pellets is controlled by chemisorption due to a good fit of the pseudo-second-order model. The best adsorption results of ACC-HAp pellets were reached with Pb(II) at pH 6, having the maximum adsorption capacity of 56 mg/g. The Pb(II) adsorption isotherm fitted more to the Langmuir model.

A successful desorption process of the Pb(II)-loaded ACC-HAp pellets was obtained by using HCl solution at pH 1.5–2. At lower pH, the mechanical strength of the pellets decreased rapidly. Due to the dissolution of the HAp coating during the desorption stage, the regeneration process had three stages—desorption, rinsing in distilled water until neutral pH, and re-coating with HAp particles. The regeneration process did not have a negative effect on repeated Pb(II) adsorption, making these pellets reusable.

Our research suggests that these pellets have the potential to be used as adsorbents. However, further research is necessary to improve their mechanical strength. Overall, the obtained results provide insight into the effectiveness of HAp-coated pellets in removing heavy metals from wastewater.

## Figures and Tables

**Figure 1 materials-16-03605-f001:**
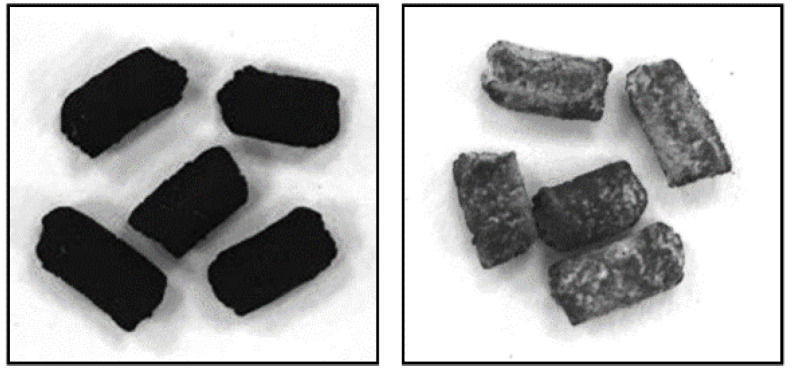
The physical appearance of ACC (to the **left**) and ACC-HAp (to the **right**).

**Figure 2 materials-16-03605-f002:**
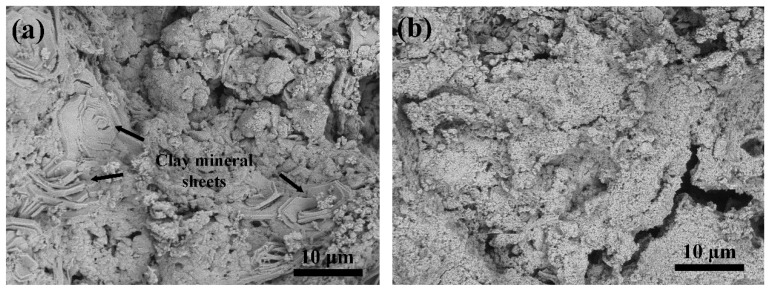
SEM images of ACC-HAp (**a**) cross-section and (**b**) surface.

**Figure 3 materials-16-03605-f003:**
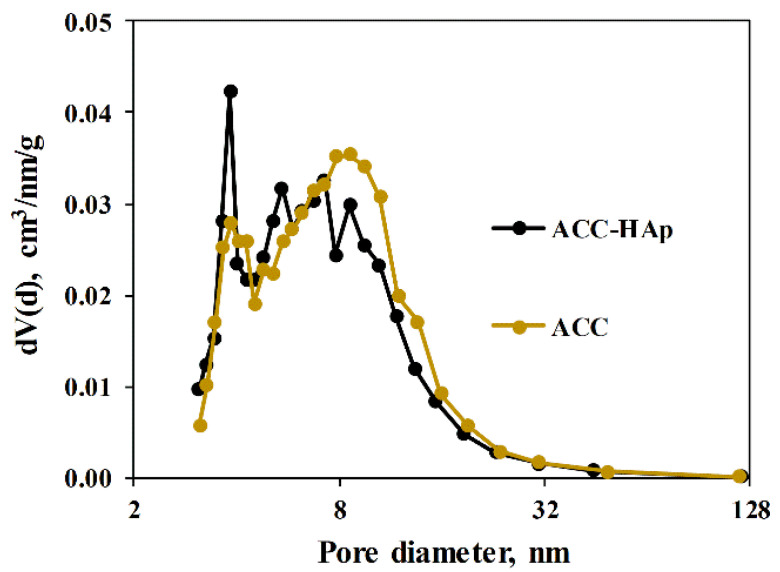
Pore size distribution of both pellets.

**Figure 4 materials-16-03605-f004:**
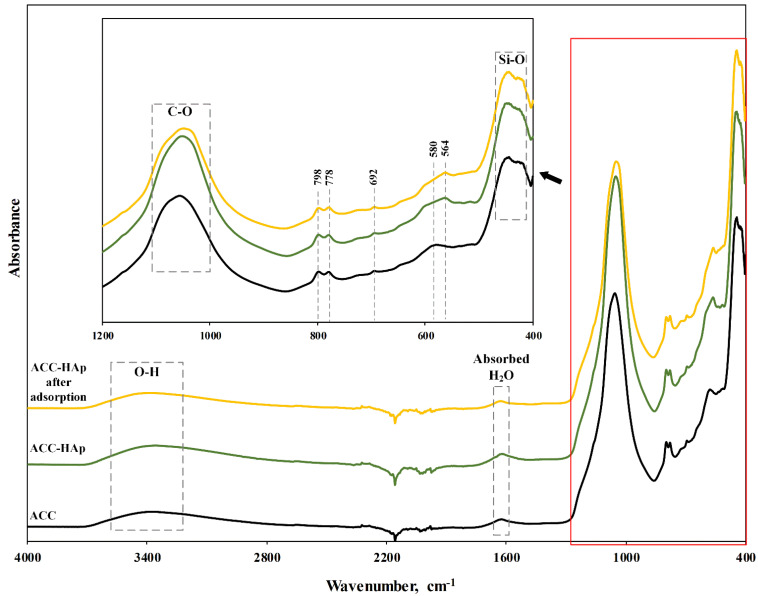
FTIR spectra of ACC pellets and ACC-HAp pellets before and after Pb(II), Cu(II), Ni(II), and Zn(II) adsorption at pH 6.

**Figure 5 materials-16-03605-f005:**
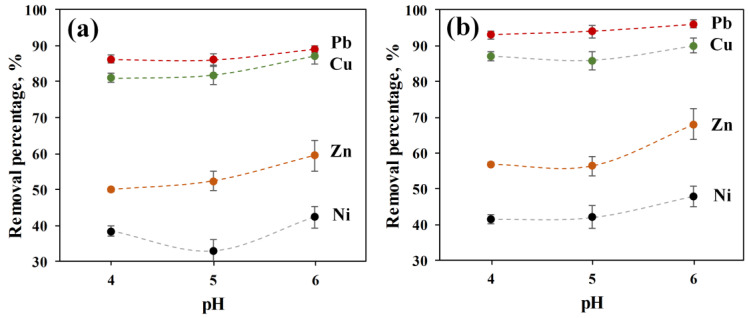
Effect of pH on the removal of Pb(II), Cu(II), Ni(II), and Zn(II) for 24 h by (**a**) ACC and (**b**) ACC-HAp pellets.

**Figure 6 materials-16-03605-f006:**
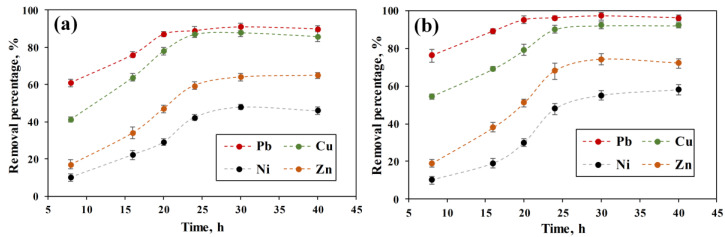
Effect of adsorption time on the removal of Pb(II), Cu(II), Ni(II), and Zn(II) at pH 6 by (**a**) ACC and (**b**) ACC-HAp pellets.

**Figure 7 materials-16-03605-f007:**
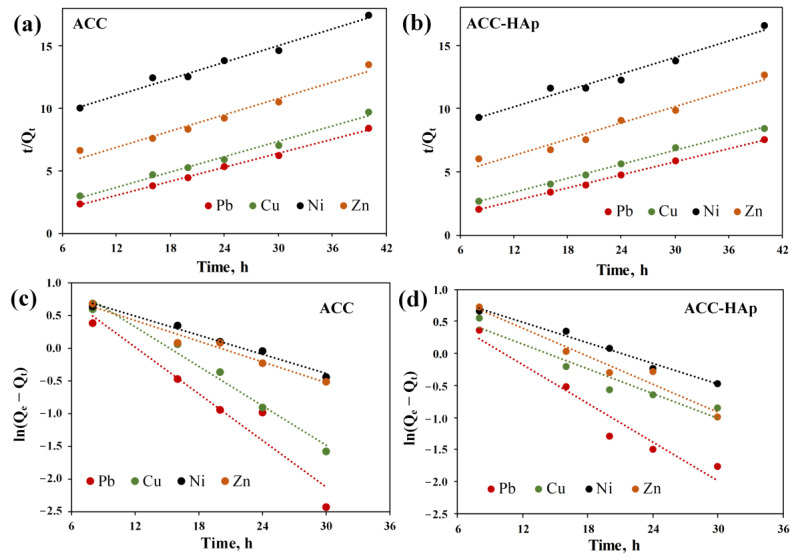
Linear fit of the PSO (**a**,**b**) and PFO (**c**,**d**) kinetic models for ACC (**a**,**c**) and ACC-HAp (**b**,**d**) pellets.

**Figure 8 materials-16-03605-f008:**
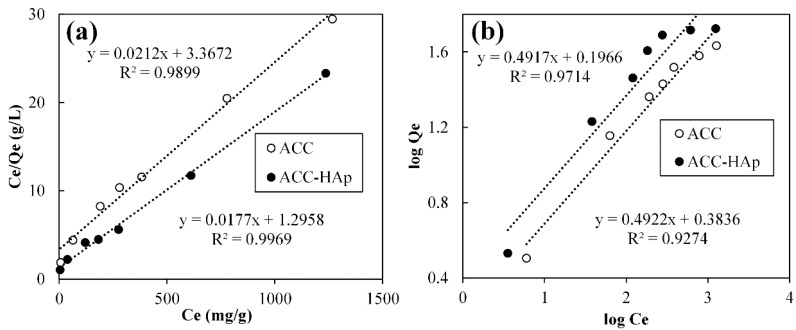
Langmuir (**a**) and Freundlich (**b**) isotherm models fitted for Pb(II) adsorption by ACC and ACC-HAp.

**Figure 9 materials-16-03605-f009:**
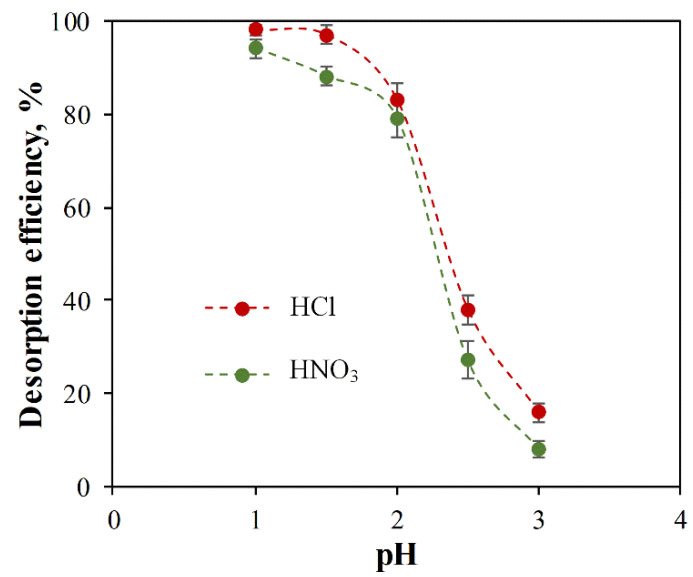
Effect of HCl and HNO_3_ at different pH values on Pb(II) desorption efficiency from ACC-HAp.

**Figure 10 materials-16-03605-f010:**
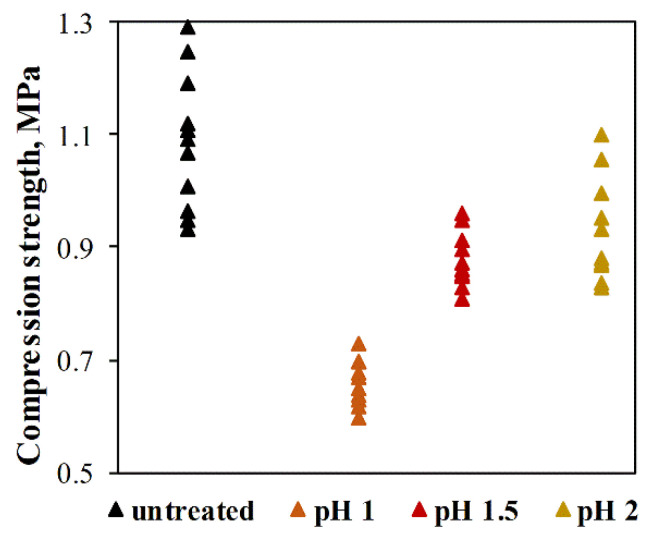
Compressive strength of ACC-HAp pellets before and after four adsorption–regeneration cycles with HCl solution at pH 1–2.

**Figure 11 materials-16-03605-f011:**
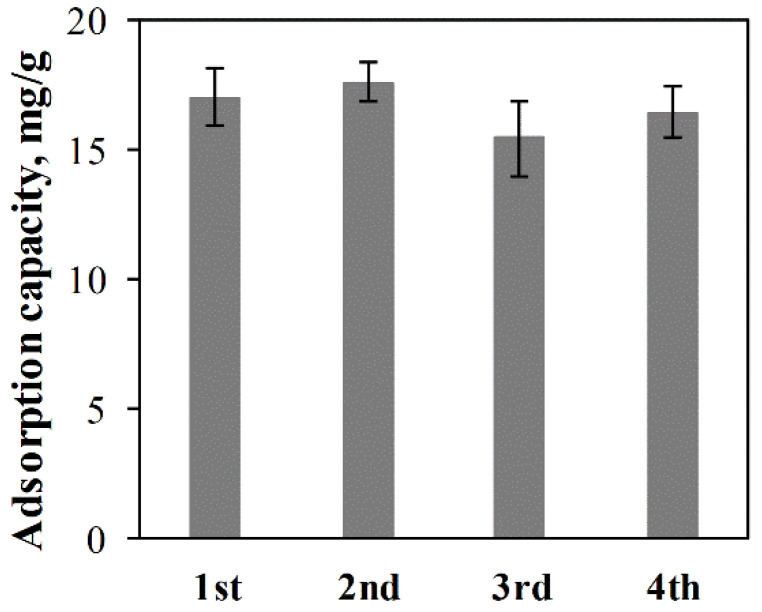
Pb(II) adsorption capacity of ACC-HAp pellets in each of four adsorption–regeneration cycles.

**Table 1 materials-16-03605-t001:** Specific surface area and total pore volume.

Samples	SSA, m^2^/g	Total Pore Volume, cm^3^/g	pH_pzc_
ACC	195 ± 8	0.41 ± 0.02	6.3 ± 0.8
ACC-HAp	188 ± 9	0.37 ± 0.03	6.5 ± 1.3

**Table 2 materials-16-03605-t002:** Kinetic parameters of Pb(II), Cu(II), Zn(II), and Ni(II) competitive adsorption.

Pellet	Ion	Experimental Data	Pseudo-First-Order			Pseudo-Second-Order		
		*Q_e, exp_* (mg/g)	k_1_ (h^−1^)	*Q_e, cal_* (mg/g)	R^2^	k_2_ (g mg^−1^ h^−1^)	*Q_e_* (mg/g)	R^2^
ACC	Pb(II)	4.6	0.27	27.7	0.9309	0.048	5.3	0.9976
	Cu(II)	4.2	0.23	34.2	0.9793	0.035	4.9	0.9888
	Zn(II)	2.9	0.12	10.9	0.9666	0.011	4.5	0.9709
	Ni(II)	2.2	0.11	11.5	0.9832	0.007	4.2	0.9853
ACC-HAp	Pb(II)	5.3	0.23	10.9	0.9411	0.22	5.7	0.9994
	Cu(II)	4.7	0.15	8.2	0.9220	0.15	5.2	0.9962
	Zn(II)	3.3	0.17	18.8	0.9580	0.19	4.6	0.9742
	Ni(II)	2.8	0.12	13.7	0.9806	0.12	4.3	0.9813

**Table 3 materials-16-03605-t003:** Langmuir and Freundlich adsorption isotherm constants for the adsorption of Pb(II).

Pellet	Langmuir Constants			Freundlich Constants		
	*q_max_* (mg/g)	K_L_ (L/mg)	R^2^	K_F_	n	R^2^
ACC	47	0.0063	0.9899	2.42	2.03	0.9274
ACC-HAp	56	0.014	0.9969	1.57	2.03	0.9714

**Table 4 materials-16-03605-t004:** Comparison with other AC adsorbents in granular and pellet forms.

Adsorbent	Pb(II) Adsorption Capacity, mg/g	Equilibrium Time, h	pH	Reference
GAC	31.1	2.25	6	[28]
GAC and HAp composite	36.9	2.25	6	[28]
GAC and HAp composite	14.41	2.5	6	[29]
GAC from hazelnut husks activated with ZnCl_2_	13.05	1	6.7	[49]
GAC from coconut shells	32.08	96	5	[50]
GAC from coconut shells	92.39	40	5	[50]
GAC modified with Na_2_S	21.88	-	5	[51]
Pellets from activated coal fly ash	45.25	72	7	[52]
Pyrolyzed pellets from phoenix tree leaf powder and bentonite mixture	71	20	5	[53]
ACC	47	20	6	This study
ACC-HAp	56	20	6	This study

## Data Availability

Not applicable.

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
