# Peer review of "Activated Carbon and Clay Pellets Coated with Hydroxyapatite for Heavy Metal Removal: Characterization, Adsorption, and Regeneration"

_materials, 2023, doi:10.3390/ma16093605_

Round 1
Reviewer 1 Report
The authors discussed on the characterization, adsorption and re-generation of ACC-HAp prepared by dip coating as compared to ACC itself. Only minor comments for improvements as follow:
1. Line 41: "high affinity towards", remove "to"
2. Line 81: check spacing after "used"
3. Line 110-111: do not bold the sentences
4. Some methods require more detailed info, such as SEM. Please include detailed methods, e.g. specimen preparation (coating used, etc), magnification, voltage, etc.
5. Line 176-177/ Figure 4: What happens if pH is increased >6, would the adsorption increase?
6. Figure 5: The gap between contact time was 4 h? How about the results for 4 h? Please include the results.
7. Table 3: standardize the equilibrium time used, either in min or hour.
8. Line 247: the sentence was not complete. Please complete it.
Author Response
- Line 41: "high affinity towards", remove "to"
The word “to” has been removed.
- Line 81: check spacing after "used"
The second spacing has been removed.
- Line 110-111: do not bold the sentences
The bold has been removed.
- Some methods require more detailed info, such as SEM. Please include detailed methods, e.g. specimen preparation (coating used, etc), magnification, voltage, etc.
Detailed information is included.
- Line 176-177/ Figure 4: What happens if pH is increased >6, would the adsorption increase?
Various studies show that for these 4 metal ions the adsorption decreases at pH > 6. Also, Pb and Cu start to precipitate in the form of hydroxide.
- Figure 5: The gap between contact time was 4 h? How about the results for 4 h? Please include the results.
The adsorption after 4 h of contact time was not conducted.
- Table 3: standardize the equilibrium time used, either in min or hour.
The equilibrium time is corrected from min to hours.
- Line 247: the sentence was not complete. Please complete it.
The sentence is completed.
Reviewer 2 Report
Manuscript “Activated carbon and clay pellets coated with hydroxyapatite for heavy metal removal: characterization, adsorption and re-generation”
Reviewer: Major Revisions
In this manuscript, authors declared the synthesis of Activated carbon and clay pellets coated with hydroxyapatite for heavy metal removal: characterization, adsorption and regeneration. Grammatical and typo-errors have been observed throughout the manuscript (yellow highlighted in the manuscript).
I would not recommend this article in Materials in its current state because of the following serious concerns about the article which should be reconsidered.
1. Page 1, abstract is not appropriate, no information is mentioned about characterization of material.
2. Infect in the whole manuscript there is no characterization except secondary characterization: BET and SEM
3. Page 1 in the introduction, lots of sentence structure errors. objectives are not clear.
Disadvantages of heavy metals and characterization of the material are not discussed in introduction paragraph.
4. Page 3 format is not same throughout in paragraph (few lines are bold).
5. Page 3 synthesis of HAp is not clear .... synthesis is not discussed in detail.
6. Conclusion is not appropriate
7. Page 12...All references are not in same format.
8. I also recommend some latest citations.
9. Language of the article needs to be polished.
10. Authors should use subscripts for the chemicals properly, errors are found in figures too (figures).
11. Figures should be colored.
12. I recommend authors to show the suggested mechanism for the removal of heavy metals in pictorial form too.
Author Response
- Page 1, abstract is not appropriate, no information is mentioned about characterization of material.
The abstract was improved. Additional information was added.
- Infect in the whole manuscript there is no characterization except secondary characterization: BET and SEM
FTIR and pHpzc results are added.
- Page 1 in the introduction, lots of sentence structure errors. objectives are not clear.
Disadvantages of heavy metals and characterization of the material are not discussed in introduction paragraph.
The errors are corrected. The objective is improved. Additional information is added in the Introduction paragraph.
- Page 3 format is not same throughout in paragraph (few lines are bold).
The bold has been removed.
- Page 3 synthesis of HAp is not clear .... synthesis is not discussed in detail.
The synthesis process is described more detailed.
- Conclusion is not appropriate
The conclusions have been corrected.
- Page 12...All references are not in same format.
The errors have been corrected.
- I also recommend some latest citations.
Several latest literature sources have been added.
- Language of the article needs to be polished.
The language of the manuscript was improved.
- Authors should use subscripts for the chemicals properly, errors are found in figures too (figures).
The errors have been corrected.
- Figures should be colored.
Some figures have been converted to colour.
- I recommend authors to show the suggested mechanism for the removal of heavy metals in pictorial form too.
Additional information on the removal mechanism has been added. Unfortunately, the pictorial form was not included due to the lack of capacity.
Reviewer 3 Report
In this study authors prepared activated carbon-containing pellets from sawdust and clays coated with hydroxyapatite. Prepared material was further invesigated as potential adsorbent of Pb(II), Cu(II), Ni(II) and Zn(II). Although the study has potential, certain shortcomings must be corrected, first of all a more comprehensive adsorption study.
1. The abstract does not reflect the main conclusions of this study.
2. In the introduction part, it is necessary to emphasize the scientific contribution and novelty of this study.
3. It is necessary to supplement the study with additional analyses, such as FTIR analysis before and after modification, as well as adsorption, it is also desirable to perform pHpzc or Zeta potential.
4. Section 3.2. Why are lower pH values not taken into account? Whether and how precipitation of metals in the form of hydroxide at pH 6 was avoided?
Kinetic models of adsorption are lacking. Comment on the impact of contact time in terms of a potential removal mechanism.
5. Include a real sample analysis.
Author Response
- The abstract does not reflect the main conclusions of this study.
The abstract was improved.
- In the introduction part, it is necessary to emphasize the scientific contribution and novelty of this study.
The scientific contribution and novelty were improved.
- It is necessary to supplement the study with additional analyses, such as FTIR analysis before and after modification, as well as adsorption, it is also desirable to perform pHpzc or Zeta potential.
FTIR analysis and pHpzc determination have been carried out and included.
- Section 3.2. Why are lower pH values not taken into account? Whether and how precipitation of metals in the form of hydroxide at pH 6 was avoided?
In many other studies on activated carbon and hydroxyapatite adsorption of heavy metals, adsorption decreases at lower pH values. There was no need to avoid the precipitation of metal hydroxides because all of the used metal ions usually start to precipitate in the form of hydroxide at pH > 6.
- Kinetic models of adsorption are lacking. Comment on the impact of contact time in terms of a potential removal mechanism.
Two kinetic models were added and described.
- Include a real sample analysis.
The aim of this research was to determine the adsorption and desorption possibilities of the pellets with standard solutions. A further study with industry (recycling plant for used lead batteries here in Latvia) is planned, where real samples will be analyzed with these pellets in a laboratory-sized adsorption column.
Round 2
Reviewer 2 Report
1. i suggest authors cite the procedure or synthesis protocol, as it is not novel(sec 2).
2. authors should first declare the selected parameters of best results for adsorption and then desorption.
3. In FTIR all peaks should be labeled (1600-2500).
4. In FTIR description we never use singlet or doublet, remove those terms.
5. In Fig 9 3 of nitric acid should be subscript as i suggested earlier.
6. still lots of grammatical errors are there along with typo.
7. why do authors characterize the material by gas sorption analyzer?
Author Response
- i suggest authors cite the procedure or synthesis protocol, as it is not novel(sec 2).
Answer: The procedure was cited.
- authors should first declare the selected parameters of best results for adsorption and then desorption.
Answer: This was corrected in the abstract and conclusion sections.
- In FTIR all peaks should be labeled (1600-2500).
Answer: The peak at around 1600 cm-1 was labeled. The absorbance between 1900-2300 cm-1 is from the diamond used in FTIR (Tolvaj, L. Traditions, anomalies, mistakes and recommendations in infrared spectrum measurement for wood. Wood Science and Technology, 2022, 56, 1819-1834). It is a background.
- In FTIR description we never use singlet or doublet, remove those terms.
Answer: The terms were removed.
- In Fig 9 3 of nitric acid should be subscript as i suggested earlier.
Answer: The figure was corrected.
- still lots of grammatical errors are there along with typo.
Answer: The noticed grammatical and typo errors were corrected.
- why do authors characterize the material by gas sorption analyzer?
Answer: It was used to determine specific surface area by BET method. “Gas sorption analyser” was replaced with “Gas sorptometer”.
Reviewer 3 Report
Authors corrected manuscript so now it can be considered for publication.
Author Response
1. Authors corrected manuscript so now it can be considered for publication.
Answer: Thank You very much for the decision.